# UV-Curable Urethane Acrylate Resin from Palm Fatty Acid Distillate

**DOI:** 10.3390/polym10121374

**Published:** 2018-12-11

**Authors:** Kim Teck Teo, Aziz Hassan, Seng Neon Gan

**Affiliations:** Chemistry Department, University of Malaya, 50603 Kuala Lumpur, Malaysia; ktteo@polymerictechnology.com (K.T.T.); ahassan@um.edu.my (A.H.)

**Keywords:** PFAD, urethane acrylate, crosslinking, UV curing, chemical resistance, T_g_, pendulum hardness

## Abstract

Palm fatty acid distillate (PFAD), is a by-product of the crude palm oil refining process. It comprises mainly of free fatty acids—around 45% palmitic and 33% oleic acids—as the major components. Ultra-violet (UV) curable urethane acrylate (UA) oligomers could be synthesized from PFAD, by the following procedure. A hydroxyl terminated macromer was first prepared by reacting PFAD with a mixture of isophthalic acid, phthalic anhydride, neopentagylcol (NPG), and pentaerythritol. The macromer was then reacted with 2-hydroxylethylacrylate (2HEA) and toluene diisocynate (TDI) to generate a resin, containing acrylate side chains for UV curable application. A series of UA resins were prepared by using 15, 25, 45, 55, and 70% of PFAD, respectively. The UA resin has M_w_ in the range of 3,200 to 27,000. They could be cured by UV irradiation at an intensity of 225 mW/cm^2^. Glass transition temperature (T_g_) of the cured film was measured by differential scanning calorimeter (DSC), and hardness of the film was determined by a pendulum hardness tester, according to American Society for Testing and Materials (ASTM)4366. The resins were used in a wood-coating application. All of the cured films showed good adhesion, hardness, and chemical resistance except for the one using the 70% PFAD, which did not cure properly.

## 1. Introduction

The coating industry has always been searching for technological improvements, to achieve greater efficiency, productivity, and lower cost. There are significant contributions that have been made by the resin industry and its ancillaries, particularly, to improve the curing rates and film properties. A coating system without a solvent would be cheaper in cost and more environment-friendly. Thus, radiation curing techniques offer many advantages, such as an energy-saving, VOC (volatile organic compounds) free, and fast curing cycle. One pack UV-curable resin system is a rapidly growing technology within the coating, adhesive, and related industries. It has found a large variety of applications, due to its high efficiency, environment-friendliness, and energy-saving nature [1,2,3]. The main components in the UV-curable formulation are the unsaturated oligomer, active diluents, and photoinitiator.

Most commercial UV-curable formulations are based on a combination of acrylic oligomers and monomers [4,5], urethane acrylate [6,7] polyester acrylate [8], and epoxy acrylate [9,10]. Properties of the UV-curable coatings are determined by the type of oligomers employed in the formulation. Currently, most available resins are petroleum-based and lacking in natural materials and sustainability.

Depletion of petroleum reserves and increasing environmental concerns have stimulated the revolution to explore materials from readily-available, renewable, and inexpensive natural resources, such as carbohydrates, oils, fats, and proteins. These renewable materials are going to play noteworthy roles in the development of a sustainable green chemistry. Oils and fats of vegetable and animal origins have been the renewable materials for the coating and ink industry.

*Elaeis guineensis* is a palm plant that produces fruit with a high oil-yield in Malaysia. It is grown commercially and has been an important crop that contributes to the economy of the country. About 90% of the palm oil is used for food, and about 10% for non-edible products, such as soaps and detergents. Traditional vegetable oils like linseed, soybean, and castor oils have been commercially used for the synthesis of alkyds, epoxides, and polyesteramides [11]. Less commonly, the cashew nut, karanja, annona squamosa, natural rubber seeds have also been investigated for producing polyurethane, polyesteramide, and alkyd resins [12,13]. Recently, there were several reports on palm oil-based resins. For example, an excellent baking enamel was produced from a water-reducible palm stearin alkyd, combined with melamine resin [14,15]. The problem of insufficient unsaturation could be overcome through the incorporation of maleic anhydride or fumaric acid into the alkyd structures, to make them UV-curable [16,17]. Further exploration of palm oil derivatives has produced many new coating resins that have various film properties [18,19,20,21]. In 2014, a US patent was granted for a palm oil-based polyurethane oligomer for use as a restorative dental material [22].

Crude palm oil consists of about 94% triglycerides, 3–5% of free fatty acids, and about 1% of other minor constituents. The palm fatty acid distillate (PFAD) is a by-product of the crude palm oil refining process. It comprises mainly of free fatty acids, with around 45% palmitic and 33% oleic acid, as the major components, and it has been used in the making of soap and animal feed, and certain oleochemicals. Small amount of Vitamin E could be extracted from the PFAD [23]. Gapor has developed a process for producing high purity (>90%) Squalene from the PFAD [24]. This valuable compound is useful in heath supplements, cosmetics, and in the pharmaceutical industry. In a recent study, the PFAD has been explored as biodiesel feed stock [25]. Many studies utilize various types of catalysts, such as SO_4_^2−^/TiO_2_-SiO_2_, and modified Zirconia compounds [25,26,27]. Malaysian palm oil refineries produce more than 750,000 MT of PFAD, annually, as a by-product, which sold at a discounted price, as compared to the RBD (Refined, Bleached and Deodorized) palm oil, at USD200–250 per metric ton, for usage in animal feed [28,29]. There are yet no reports of its application in coating.

Similar to any other coating technologies, wood coating is one of the important segments in the coating industry that focuses on developing more environment-friendly resins, in comparison to traditional solvent-borne coating system. UV-curable resins offer less VOC, lower energy in the curing process, and shorter curing time. The worldwide market for Wood Coating is expected to grow at a compound annual growth rate of roughly 5.9%, over the next five years, and will reach 11,700 million US$, in 2023, from 8740 million US$, in 2017 [30]. Malaysia, as one of the ten largest furniture exporters, has provided a very good economic base for the wood coating industry development. Modernization of architectural designs, including furniture, has driven the increasing need for improved aesthetic appeal of furniture and other wooden products, making wood coating a very important part of the woodworking industry. In addition, the material needs to be protected against mechanical, physical, and chemical attack. The available technologies include waterborne coatings, high-solids, and UV-cured coatings. There is a noticeable move from the solvent-borne coatings to solvent-free or solvent-reduced materials, driven by the environmental and regulatory demands. Other than conventional waterborne technologies where acrylate/vinyl emulsions are the main binder used, the solvent-borne technology has involved many types of alkyd resin as the main binder in the three, major coating systems, such as acid catalyzed, nitrocellulose, and polyurethane wood coating system. In a high-solid system, UV-curing is the current trend and the focus of development is to achieve 100% solidity, in a cured coating.

We have investigated the use of PFAD in wood-coating resins. To fulfil the current technology trend of low VOC or no VOC, PFAD-based resins should be extended into high solid or UV-curable resins for wood-coating applications. This project has aimed to synthesize PFAD-based urethane acrylates with different PFAD contents. The urethane acrylates (UA) were investigated as UV-curable resins for wood coating applications.

## 2. The Experimental Method

### 2.1. Materials

The PFAD was a kind gift from (Sime Darby Group, Kuala Lumpur, Malaysia). This commercial PFAD was used without further treatment. 2,2-dimethyl-1,3-propanediol (neopentaglycol, NPG) was purchased from (LG Chemicals, Yeosu, Korea), 2,2-bis(hydroxylmethyl)1,3-propanediol (Pentaerythritol) from (Perstop AB, Malmo, Sweden), phthalic anhydride from (Nanya Plastic, Kaohsiung, Taiwan), purified isophthalic acid from (MGC Chemicals, Tokyo, Japan), mono-butyl tin oxide from (Arkema, Colombes, France), toluene diisocyanate (80/20 TDI) from (Covestro AG, Leverkusen, Germany), 2-hydroxy ethyl acrylate (2-HEA) from (BASF, Shah Alam, Malaysia), xylene from (Exxonmobil, Leatherhead, UK), Tripropylene glycol diacarylate (TPGDA), triphenyl phosphine (TPP) and 4-methoxy phenol from (Sigma Aldric Chemicals, Saint Louis, MS, USA), Speedcure 73 (2-hydroxy-2-methylpropiophenone) from (Lambson, Wetherby, UK). Tetrahydrofuran (THF) from (Thermo Fisher Scientific, Waltham, MA, USA). All Chemicals were used as received.

### 2.2. Synthesis

#### 2.2.1. Synthesis of the PFAD Hydroxyl-Terminated Macromer

A hydroxyl-terminated macromer was first synthesized by one batch reaction of PFAD, isophthalic acid, phthalic anhydride, neopentagylcol, and pentaerythritol. The procedure is a modified process that had been described elsewhere [30].

PEAD and Neopentaglycol were charged into a 2 L round-bottom flask, fitted with a decanter, and stirrer. The mixture was heated slowly to 100 °C, until all of the material melted, and subsequently pentaerythritol, isophthalic acid, phthalic anhydride, and mono-butyl tin oxide were added. Maintaining stirring at 250 rpm, the temperature was raised, gradually, and water from the condensation reactions that evolved above 180 °C, was trapped and removed from the decanter. Temperature was raised and maintained at 235–240 °C. Progress of the reaction was monitored by measuring the acid value and viscosity of a small sample taken from the reactor, at different reaction intervals. The sample was diluted with xylene to an 80% solid content, before its viscosity was measured by the Brookefield viscometer (model DVI). Acid value was determined by titrating a known weight of the sample with a standardized NaOH solution, according to the ASTM (American Society for Testing and Materials) D4662-93 Method. When the acid value had fallen below 10 mgKOH/g, heating was turned off and allowed to cool down to 100 °C, before the PFAD macromer was transferred to a storage container and diluted with TPGDA, to a 70% solid content and kept for further use. Five macromers were synthesized with a PFAD content of 15%, 25%, 45%, 55%, and 70%, respectively, the recipes of the reactions are as shown in Table 1.

#### 2.2.2. PFAD Urethane Acrylate Oligomer (UA)

The PFAD urethane acrylate oligomer was synthesized via the reactions of the PFAD macromers (M1–M5) with 2HEA and TDI. The procedure is as follows. The TDI (95.9 g) was charged into a four-necked flask equipped with a mechanical stirrer, dropping funnel, thermometer, and a nitrogen inlet. The flask was thermostated to 35 °C in a water bath. The 2HEA (63.4 g) and 0.1 g of the 4-methoxyphenol were mixed in a separate flask and the mixture dosed into the flask, dropwise, over a period of 2 h. The progress of the reaction could be checked by both the FTIR spectrum and by determining the isocyanate value through titration. When the isocyanate content had reduced to half of the initial value, the –OH peak in the FTIR spectrum would have disappeared, and the specified amount of the PFAD macromer was charged into the reaction mixture. The temperature was raised to 97 °C to accelerate the reaction, which was stopped when the residual isocyanate content had dropped to less than 0.05%. Then 64.0 g of the TPGDA was added and the final resin was ready for further evaluation. Table 2 summarizes the compositions of the urethane acrylic oligomers UA1, UA2, UA3, UA4, and UA5 prepared from the PFAD macromers M1, M2, M3, M4, and M5, respectively.

#### 2.2.3. UV-Curable Formulation

The UV curable formulations were prepared by mixing the urethane acrylic oligomer (UA) at 80.00% (*w*/*w*), Speedcure 73 at 3.00% (*w*/*w*), and 17.00% (*w*/*w*) TPGDA.

### 2.3. Characterization

#### 2.3.1. Molecular Weight Measurement

Molecular weights of the samples were measured by a Gel permeation chromatography (GPC) (Shimadzu, Kyoto, Japan) with tetrahydrofuran as the mobile phase (set at 1.0 mL min^−1^). Monodisperse polystyrene standards were used for the calibration of the column.

#### 2.3.2. FTIR Analysis

FTIR spectra were recorded by using the ATR technique on a Perkim Elmer Spectrum Two FTIR spectrometer (Perkin Elmer, Waltham, MA, UAS). The polymer was coated directly onto the ATR diamond plate and the spectrum was recorded.

#### 2.3.3. Viscosity Measurement

A Brookefield viscometer (model DVI) was used to measure the sample dynamic viscosity at a temperature of 25 °C. A sample of macromer was diluted to 70% of the oligomer content, with TPGDA, before viscosity measurement.

#### 2.3.4. Glass Transition Temperature

To determine the T_g_ of the resin and the cured film, samples were encapsulated in a 40 μL aluminum pan and analyzed using the Shimadzu DSC (model DSC-60), at a heating rate of 20 °C min^−1^.

#### 2.3.5. Pendulum Hardness

The hardness of the UV-cured film was measured with the Sheen Konig pendulum hardness tester (TQCSHEEN, Capelle aan den IJssel, Netherlands), according to ATSM 4366.

#### 2.3.6. Acid Value, Isocyanate Value, and Hydroxyl Value

The resin acidity (acid value), isocyanate value, and hydroxyl value were determined by the standard titration method, according to the ASTM D4662-93, ISO 14896/3, and ASTM D1957-86 (2001), respectively.

#### 2.3.7. Gloss Measurement

The gloss of the cured film was measured with the BYK tri-gloss meter, according to the ASTM D523, for non-metallic substrate. The gloss level was reported as the gloss unit (GU).

#### 2.3.8. Adhesion

The film adhesion was performed according to the ASTM D3359-09 test method B. The adhesion performance was rated as shown in Table 3.

#### 2.3.9. Pencil Hardness

The hardness of the cured film was measured by the pencil hardness, according to ASTM D3363. The pencil hardness scale is as follow, where 6B denotes the softest and 6H being the hardest: Pencil Hardness Scale: 6B-5B-4B-3B-2B-B-HB-F-H-2H-3H-4H-5H-6H.

### 2.4. Monitoring of the UV Curing

UV curing was carried out by casting the formulated UA resin, using a bar coater of 20 µm, onto a glass panel. The film was then irradiated with UV light at a distance of 10 cm, from the source, for the specified exposure time, for 5 to 60 s. The UV light was provided by a mercury vapor lamp with an intensity of 225 mW/cm^2^, at a UV wavelength of 254–365 nm. The extent of cure at each instant, was correlated to the hardness and the T_g_ of the film.

### 2.5. Chemical Resistance of the Cured Film

The cured films were subject to the wood-coating standards of the chemical resistance test. The coating was applied on a rubber wood panel and cured by irradiation, with a UV light, for 60 s. The cured panel was conditioned at room temperature, for 1 h, before carrying out the chemical resistance test. Ten milliliter of chemical reagent was dropped onto the film surface and allowed to stand for 24 h; to minimize the evaporation of the reagent, it was covered with a plastic cap. The reagent was then wiped off with a dry clean cloth, and the surface was then dried for 15 min, before making observations and reporting. The reagents commonly applied by the Industrial Wood-Coating Standard include coffee, tea, dish washing solution, acetone, cooking oil, 1% ethanol aqueous solution (*wt*/*wt*), vinegar (8% acetic acid, *v*/*v*), 10% ammonia solution, 5% sodium hydroxide (NaOH) solution (*wt*/*v*), and 5% hydrochloric (HCl) solution (*w*/*v*).

## 3. Results and Discussions

### 3.1. Synthesis of the PFAD Macromer

The hydroxyl terminated PFAD macromere was synthesized from the reactions between the PFAD, the neopentylglycol, the pentaerythritol, the isophthalic acid, and the phthalic anhydride. A plausible structure of the macromer is shown in Scheme 1.

Figure 1 shows the change in acid values, with reaction time, during the synthesis of the macromers. With reference to Table 1, the PFAD content in an increasing order is M1 < M2 < M3 < M4 < M5, and all the macromers showed a similar reaction trend, being fast at the initial stage from 120–360 min, and reached a constant AV, after 420 min. Between 180–360 min, the acid value of all macromers decreased at comparable rates, all of them reached an acid value <10 mgKOH/g.

The M_w_ of each macromer is shown in Table 4. In general, molecular weight of the alkyd resin is known to be affected by its oil-length [31]. In this case, from M1 to M3, with %PFAD increases from 15% to 45%, the M_w_ showed an increasing trend, from 2,400 to 8,000. However, for the M4 and the M5, at %PFAD of 55% and 70%, respectively, the M_w_ had dropped to around 2,700. In this series of study, the macromer had attained the highest M_w_, with the formulation of M3, containing 45% PFAD, at 8,000.

The PFAD consisted of fatty acids which were mono-functional carboxylic acids. In general, the short oil-length alkyd had a fatty acid content of 20–40%, and the polymerization was mainly controlled by the polyhydric alcohol and the polybasic acid used. When the fatty content was in the range of 40–55%, based on the Patton alkyd formulation calculations [31], the polyhydric alcohol and the polybasic acid was 60–45%, in the formulation, to achieve the optimum performance, in terms of molecular weight. However, when the fatty acid content exceeded 55%, the polyfunctional reactants was much reduced (<45%), in the alkyd formulation. This could result in a termination reaction by the mono-functional fatty acid, to become predominant during the esterification process, and could produce a lower molecular weight polymer.

The FTIR spectra of the macromers are shown in Figure 2 and Figure 3. The hydroxyl groups of the polyols are seen as –OH, stretching at 3500 cm^−1^. The peaks at 2852–2956 cm^−1^ were associated with the symmetric and asymmetric stretchings of the C–H, the aliphatic –CH_2_, and the –CH_3_. Other bands included the –C=O, at 1722 cm^−1^, and the C–O–C stretching, at 1120–1230 cm^−1^. In Figure 2, the M1, M2, and M3, peaks of the –C=C– of acrylate, at 1409 and 809 cm^−1^, were contributed by the diluents TPGDA. The dilution of these three macromers were necessary as the neat macromers showed an extremely high viscosity and were not possible for the subsequent grafting process. Figure 3 shows the spectra of M4 and M5, which were not diluted with TPGDA, hence the acrylate peak was not observed. The characteristic properties of the PFAD macromer are summarized in Table 5.

### 3.2. PFAD Urethane Acrylate (UA)

The PFAD macromer was reacted with TDI and 2-HEA to produce the UA resin. TDI contained –NCO at the ortho and para positions, and the para-NCO was eight times more reactive than the ortho-NCO, at temperatures below 30 °C [32,33]. The adduct of the TDI and the 2HEA was first produced at temperatures below 30 °C. The consumption of NCO was monitored through the reduction of peak at 2270 cm^−1^, in the FTIR spectrum. The plausible reaction path of the PFAD urethane acrylate formation is shown in Scheme 2.

Figure 4 shows the change in %NCO, with reaction time, during the synthesis of the PFAD-based urethane acrylate. The initial NCO of 49.2% had reduced, gradually, and reached 0.05% at the end of the polymerization process. At time intervals of 120 min, UA1 and UA2 showed a slower rate of formation, as compared to the UA3, UA4, and UA5, presumably due to the effect of viscosity. The high viscosities of M1 and M2 had reduced the mobility of TDI adduct, during the grafting process, and resulted in a slower reaction rate, to form UA1 and UA2, respectively. The low viscosities of M3, M4, and M5 provided a better mobility medium for the TDI adduct to react, as clearly reflected in the fast reduction rate of the NCO value, particularly from a reaction time of 60–240 min.

Figure 5 shows the FTIR spectra of the macromer M4 and the final urethane acrylate of UA4. The shift of peak at 3500 to 3350 cm^−1^, was due to the reaction of –OH and –NCO, to form the urethane linkage. The N–H deformation at 1531 cm^−1^ and the acrylate double bond at 810 cm^−1^, were observed on the UA4 but not on the M4.

Figure 6 shows the FTIR spectra of all the UA resins. The shift of peak at 3500 cm^−1^ to 3350 cm^−1^ and the N–H deformation at 1532cm^−1^ confirmed the formation of the urethane linkage, after the grafting reaction.

The molecular weight of a polymer relative to polystyrene standards were determined by the GPC. As expected, the M_w_ of the UA was higher than its macromer. The increase was due to the further reactions of the macromer with a TDI and a 2-HEA adduct. Properties of the UA resins are summarized in Table 6.

### 3.3. T_g_ of the UV-Cured Film by the DSC Method

A clear coat was prepared by applying the resin onto the glass panel by a bar coater, to achieve a thickness of 25 µm and was irradiated with a UV lamp, at specified durations. T_g_ of the cured-film was determined by DSC.

Figure 7 shows the change in the T_g_ of the film, at different UV curing times. Generally, all formulations show rapid photopolymerization at 5 to 20 s UV exposure. Subsequently, curing has slowed down as the acrylate double bond was used up. The curing reaction was accompanied by an increase in viscosity, which would reduce the mobility of the reactant with the effect of reducing the rate of reaction. As shown in Figure 7, the %PFAD in the UA has a great influence on the T_g_ of the cured-film. Generally, the T_g_ was reduced with a higher %PFAD. The UA5 film of showed the lowest T_g_ (29 °C), after 60 s of UV irradiation. The UA3 film had a higher T_g_ value, among the UA2 to UA5. In case of the UA1, its highly reactive polyester backbone had a dominant contribution to the T_g_, thus, the UA1 had attained the highest T_g_ (63 °C), as compared to the other four.

### 3.4. Pendulum Hardness of the Cured Film

The film at different stages of UV curing was subjected to the pendulum hardness test by using the Sheen Konig pendulum tester, according to the ASTM 4366. The results are shown in Figure 8.

Only four resins (UA1 to UA4) had films with a measurable hardness during the UV-curing from 5–60 s. The UA5 film was very soft, as reflected by its low-film T_g_ (<30 °C). In general, the hardness, as well as the T_g_ of the cured film, had increased during the photo-induced crosslinking process. They were mainly dependent on the cross-link density and the nature of the resin. Similar to the T_g_ property, the hardness of the film was influenced by the %PFAD. Thus, UA1 with the lowest %PFAD and more reactive polyester backbone had showed a good hardness development, as compared to the others and achieved the final pendulum hardness of 118 s. As %PFAD increased, the UA3 showed optimum cured film properties and achieved the final film hardness of 111 s. On the other hand, UA2 and UA4 showed quite similar hardness, after 5 s of UV curing time, but subsequently the hardness development of UA2 increased faster than the UA4, from 10 to 30 s. At 30 s, UA2 and UA4 had achieved pendulum hardness of 77 s and 49 s, respectively, and the final pendulum hardness 88 s and 55 s, respectively. Presumably the difference was due to the M_w_, where UA2 had a higher M_w_ of 7500 dalton, as compared to UA4, at M_w_ of 3800 dalton.

### 3.5. Chemical Resistance of the UV-Cured Film

All wood panels were coated with the formulation described in Section 2.2.3. The initial cured film properties and appearance are summarized in Table 7 and the final properties and appearance are showed in Table 8 and Table 9.

Results from Table 7 showed that all cured films had good physical properties except the UA5 that remained tacky. UA1–4 showed a glossy appearance with gloss measurements ranging from 93–97 GU. All cured films had excellent adhesion onto the wood panel, with a recorded grade of 5B. In the pencil hardness scale, UA1 and UA3 showed a hardness of H, followed by UA2 of F and UA4 of HB.

In Table 8, the cured films of UA1, UA2, and UA3 exhibited a good chemical resistance with an unchanged film appearance, after 24 h aging, with almost all the chemicals. Generally, the films showed a poor alkaline-resistance. Fading and whitish marks were observed on the cured film when testing was done with a 5% NaOH solution. UA4 (55% PFAD) showed poorer film properties, as compared to UA1, UA2, and UA3. Fading and whitish marks were observed on the wood panels, for tests using the dish-washing solution and a 5% NaOH solution. The UA4-cured film showed slight fading in vinegar, 10% ammonia solution, and 5% HCL solution aging test. UA5(70% PFAD) formed soft and tacky film and was excluded from the tests mentioned in Table 8 and Table 9.

Table 9 shows the results of adhesion and pencil hardness tests performed on the all-cured film, after 24 h chemical aging test. For the films of UA1, UA2, and UA3, results for crosshatch adhesion and pencil hardness remained perfect, except for those after the 5% NaOH solution aging. In the case of the pencil hardness test, films of the UA1, UA2, and UA3 were not affected after the chemical aging, except for the 5% NaOH solution. Their hardness remains unchanged at the initial values of H, F, and H, for the UA1, UA2, and UA3, respectively. The UA4 film showed a significant reduction in performance, especially in the 5% NaOH and in the dish-washing solution. The cured film adhesion had reduced to grade 2B and 3B and a hardness of 3B and 2B were recorded, after aging with the 5% NaOH solution and the dish-washing solution, respectively. A slight reduction of film properties was observed in vinegar, 10% ammonia solution, and a 5% HCl solution, for the UA4. The crosshatch adhesion had dropped one grade lower than its initial value and the pencil hardness was recorded at B, for these three chemical aging tests.

Figure 9 shows the gloss retention of the films, after 24 h of chemical aging. In general, all cured films that showed a good chemical resistance also showed a very high gloss retention. As expected, the films that had performed badly in the 5% NaOH solution test, showed a lower gloss retention ranging from the 68–81%. Except in tea, acetone, cooking oil, and the 5% HCl solution aging test, the UA4 showed the lowest gloss retention in the dish-wash solution, vinegar, the 10% ammonia solution, and the 5% NaOH solution aging, as compared to the other three resins. As shown in Figure 9, UA2 showed the highest gloss retention for all chemical aging tests, followed by the UA1 and UA3. The high gloss retention achieved by the UA resins (>90%) is highly desirable for industrial wood-coating applications.

## 4. Conclusions

Five hydroxyl terminated macromers can be synthesized from 15–70% PFAD to react with a mixture of isophthalic acid, phthalic anhydride, neopentagylcol (NPG), and pentaerythritol. Each macromer was then reacted with 2-hydroxylethylacrylate (2HEA) and a toluene diisocynate (TDI) adduct to generate a urethane acrylic (UA) resin containing acrylate side chains that can be cured by UV irradiation. The content of the PFAD had a significant influence on the final film properties. The urethane acrylic oligomers with 15–55% PFAD, in the macromers, could be UV-cured to form hard films with a T_g_ of 35–62 °C. UA5 that is made from M5 containing 70% PFAD in the macromere, cured to form a soft film with T_g_ < 30 °C. Thus, it can be concluded that urethane acrylate oligomer formulated with macromer containing 70%, and above, PFAD would not form a useful UV-cured film. Therefore, to obtain a PFAD-based urethane acrylate oligomer with optimum performance, the synthesis of the PFAD macromere should be within the range of 40–55% PFAD. From the UV-cured film, the chemical and physical properties of the oligomer formulated with macromere containing 45% PFAD, could be useful and valuable bio-resources for UV-cured wood-coating applications.

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
