# Peer review of "UV-Curable Urethane Acrylate Resin from Palm Fatty Acid Distillate"

_polymers, 2018, doi:10.3390/polym10121374_

Round 1
Reviewer 1 Report
The authors reported a systematic work about UV curable urethane acrylate resin based on Palm Fatty Acid distillate. A wide spectrum of characterizations have been conducted to demonstrate their working flow on the functional resins. However, the main weakness of the work is clarification of the polymer structure. Is it physical mixture can do the same job? Only FTIR is not enough to convince audience what authors have synthesized. It is also important to show the exact structure before and after the UV curing. So I suggest author to add more measurement on the polymer and polymer precursors, for example solid C NMR.
Author Response
Comment of reviewer 1
The authors reported a systematic work about UV curable urethane acrylate resin based on Palm Fatty Acid distillate. A wide spectrum of characterizations have been conducted to demonstrate their working flow on the functional resins. However, the main weakness of the work is clarification of the polymer structure. Is it physical mixture can do the same job? Only FTIR is not enough to convince audience what authors have synthesized. It is also important to show the exact structure before and after the UV curing. So I suggest author to add more measurement on the polymer and polymer precursors, for example solid C NMR.
We thank the reviewer for his kind comments. We have also obtained assistance from another colleague to check the English, and have made corrections on grammatical errors and rewritten many parts of the manuscript. The corrections are highlighted in yellow. Please see the revised manuscript.
As there are more than three different monomers in the reactions to form the PFAD macromers and then the urethane acrylate oligomers, there are many possible structures from the combination of the functional groups. Adopting the usual approach (some as many other authors on similar related subjects), the reaction schemes 1 & 2 mean to represent the plausible structures on how the reactive functional groups were involved.
The FTIR spectra in Figures 2 & 3 highlighted the presence of -OH of the hydroxyl groups and other characteristic peaks from other monomers that have been incorporated during the reactions. Spectra in Figure 5 shows the changes when the macromer M4 is converted to the final urethane acrylate of UA4. The shift of peak at 3500 to 3350 cm-1 was due to the reaction of -OH and -NCO to form the urethane linkage. The -N-H deformation at 1531 cm-1 and acrylate double bond at 810 cm-1 were observed on UA4 but not on M4.
UV curing involves crosslinking reactions at the unsaturation sites. Its occurrence leads to changes in Tg, hardness in the films, solvent resistant, etc. Measurements to support the application as wood coating have been reported.
We did not carry out C-NMR study.

Reviewer 2 Report
Review of the manuscript entitled ‘UV Curable Urethane Acrylate Resin from Palm Fatty Acid Distillate’
The manuscript deals with the synthesis of urethane acrylate with different Palm fatty acid distillate (PFAD) content. The urethane acrylates (UA) were investigated as UV curable resins for wood coating applications.
The manuscript must be checked to correct the English grammar.
- It was not detailed in the introduction why authors have chosen wood coating applications. Why this material is better for this application?
- What is the wavelength of the UV lamp ?
- What is the rubber wood panel? Could it be apply to any woods?
- Figure 1 what is M1. M2, M3 and M4?
- Could you explain the tab 4. Why do you observe this value for M3?
- The legend of all figures is on the top of the x and y axis values.
- Figure 7 there are 5 curves and only 4 legends. What is the ‘+’?
- Figure 8 same problem. No unity on the y axis
- The conclusion must be written again with a clear expectation of the material and the final result. You must also conclude on the better material to be used.
Author Response
Point 1 : The manuscript must be checked to correct the English grammar.
We thank the reviewer for his kind comments. We have asked another colleague to check the English, and have made corrections on grammatical errors and rewritten many parts of the manuscript. The corrections are highlighted in yellow. Please see the attached revised manuscript.
Point 2 - It was not detailed in the introduction why authors have chosen wood coating applications. Why this material is better for this application?
A paragraph on wood coating is added in Page 3.Similarly, to any other coating technologies, wood coating is one of the important segments in coating industry that focused on developing more environment-friendly resins in comparison to traditional solvent-borne coating system. UV-curable resins have offered less VOC, lower energy in the curing process, and shorter curing time. The worldwide market for Wood Coating is expected to grow at a compound annual growth rate of roughly 5.9% over the next five years, and will reach 11700 million US$ in 2023 from 8740 million US$ in 2017 [30]. Malaysian as one of the ten largest furniture exporters has provided a very good economic base for wood coating industry development. Modernization of architectural design including furniture has driven the increasing need for improved aesthetic appeal of furniture and other wooden products, making wood coating a very important part of the woodworking industry. In addition, the material needs to be protected against mechanical, physical and chemical attack. The available technologies include waterborne coatings, high-solids and UV cured coatings. There is a noticeable move from solvent borne coatings to solvent-free or solvent-reduced materials, driven by environmental and regulatory demands. Other than conventional waterborne technologies where acrylate/vinyl emulsions are the main binder used. The solvent borne technology has involved many types of alkyd resin as main binder in the three major coating system such as acid catalysed, nitrocellulose and polyurethane wood coating system. In high solid system, UV cured is the current trend and the focus of development is to achieved 100% solid in cured coating.
Point 3 - What is the wavelength of the UV lamp ?
Sorry for the omission. We have explained it on page 9. The UV light was provided by a mercury vapour lamp with intensity of 225 mW/cm2 at UV wavelength from 254-365 nm.
Point 4 - What is the rubber wood panel? Could it be apply to any woods?
Currently the rubber wood is the most widely used material for furniture in Malaysia. It is harvested from the rubber plantation during the replanting process. The coating can be applied to other wood surfaces.
Point 5 - Figure 1 what is M1. M2, M3 and M4?
M1-M4 refer to the PFAD hydroxyl terminated macromers. The compositions of the synthesis of these macromers are given in Table 1 (page 6)
Point 6 - Could you explain the tab 4. Why do you observe this value for M3?
Page 12, Table 4.
The discussion on Table 4 is given in Pages 12-13. We have added one paragraph.
The Mw of each macromer is shown in Table 4. In general, molecular weight of alkyd resin is known to be affected by its oil-length [31]. In this case, from M1 to M3, %PFAD increases from 15 to 45%, Mw has shown an increasing trend, from 2400 to 8000. However for M4 and M5, at %PFAD of 55 and 70% respectively, Mw has dropped to around 2700. In this series of study, the macromer has attained the highest Mw with formulation of M3, containing 45% PFAD at 8000.
PFAD consist of fatty acids which are mono functional carboxylic acids. In general, short oil length alkyd has fatty acid content of 20-40%, and the polymerization was mainly control by polyhydric alcohol and polybasic acid used. When the fatty content is in the range of 40-55%, based on Patton alkyd formulation calculations [31], the polyhydric alcohol and polybasic acid will be 60-45% in the formulation to achieve the optimum performance in term of molecular weight. However, when fatty acid content exceeds 55%, the polyfunctional reactants will be very much reduced (<45%) in the alkyd formulation. This could result in termination reaction by the mono functional fatty acid to become predominant during the esterification process and produce lower molecular weight polymer.
Point 7- The legend of all figures is on the top of the x and y axis values.
We have made corrections to follow the recommended format of the Journal. The legend of figure is moved to the bottom.
Point 8 - Figure 7 there are 5 curves and only 4 legends. What is the ‘+’?
The corrections have been made. ‘+’ for UA5
Point 9- Figure 8 same problem. No unity on the y axis
The corrections have been made. Pendulum hardness of UA5 film could not be measured
Point 10 - The conclusion must be written again with a clear expectation of the material and the final result. You must also conclude on the better material to be used.
The conclusion has been rewritten.
Five hydroxyl terminated macromers can be synthesized from 15-70% PFAD to react with a mixture of isophthalic acid, phthalic anhydride, neopentagylcol (NPG) and pentaerythritol. Each macromer is then reacted with 2-hydroxylethylacrylate (2HEA) and toluene diisocynate (TDI) adduct to generate a urethane acrylic (UA) resin containing acrylate side chains that can be cured by UV irradiation. The content of PFAD has significant influence in the final film properties. The urethane acrylic oligomers with 15-55% PFAD in the macromers could be UV cured to form hard films with Tg of 35-62ºC. UA5 that is made from M5 containing 70% PFAD in macromere, cured to form a soft film with Tg < 30ºC. Thus it can be concluded that urethane acrylate oligomer formulated with macromer containing 70% and above of PFAD would not form a useful UV cured film. Therefore, to obtain a PFAD-based urethane acrylate oligomer with optimum performance, the synthesis of the PFAD macromere should be within the range of 40-55% PFDA. From the UV cured film chemical and physical properties of oligomer formulated with macromere containg 45% PFAD could be useful and valuable bio-resources for UV cured wood coating application.

Round 2
Reviewer 2 Report
of for publication